

# Early Holocene morphological variation in hunter-gatherer hands and feet

Kara C. Hoover[1] and J. Colette Berbesque[2]

[1] Department of Anthropology and Department of Biochemistry and Molecular Biology, University of Alaska, Fairbanks, AK, United States of America
[2] Centre for Research in Evolutionary, Social and Inter-Disciplinary Anthropology, University of Roehampton, London, United Kingdom

## ABSTRACT

**Background**. The Windover mortuary pond dates to the Early Archaic period (6,800–5,200 years ago) and constitutes one of the earliest archaeological sites with intact and well-preserved human remains in North America. Unlike many prehistoric egalitarian hunter-gatherers, the Windover people may not have practiced a sex-based division of labor; rather, they may have shared the load. We explore how mobility and subsistence, as reconstructed from archaeological data, influenced hand and foot bone morphology at Windover.

**Methods**. We took length and width measurements on four carpal bones, four tarsal bones, and load-bearing tarsal areas (calcaneus load arm, trochlea of the talus). We analyzed lateralization using side differences in raw length and width measurements. For other hypothesis testing, we used log transformed length-width ratios to mitigate the confounding effects of sexual dimorphism and trait size variation; we tested between-sex differences in weight-bearing (rear foot) and shock-absorbing (mid foot) tarsal bones and between-sex differences in carpal bones.

**Results**. We identified no significant between-sex differences in rear and midfoot areas, suggesting similar biomechanical stresses. We identified no significant between-sex differences in carpal bones but the test was under-powered due to small sample sizes. Finally, despite widespread behavioral evidence on contemporary populations for human hand and foot lateralization, we found no evidence of either handedness or footedness.

**Discussion**. The lack evidence for footedness was expected due its minimal impact on walking gait but the lack of evidence for handedness was surprising given that ethnographic studies have shown strong handedness in hunter-gatherers during tool and goods manufacture. The reconstructed activity patterns suggested both sexes engaged in heavy load carrying and a shared division of labor. Our results support previous findings—both sexes had stronger weight-bearing bones. Male shock-absorbing bones exhibited a trend towards greater relative width (suggesting greater comparative biomechanical stress) than females which may reflect the typical pattern of male hunter-gatherers engaging in walking greater distances at higher speeds than females. While there were no significant between-sex differences in carpal bones (supporting a shared work load model), females exhibited greater variation in index values, which may reflect a greater variety of and specialization in tasks compared to males. Because carpals and tarsals are so well-preserved at archaeological sites, we had surmised they might be useful proxies for activity in the absence of well-preserved long bones. Tarsals provide a stronger signal of past activity and may be useful in the absence of, or in addition to,

Corresponding author
Kara C. Hoover, kchoover@alaska.edu

preferred bones. Carpals, however, may not be useful as the effect size of biomechanical stress (in this study at least) is low and would require larger samples than may be possible at archaeological sites.

# INTRODUCTION

A sex-based division of labor is seen across most human societies for the majority of evolutionary history and contributes to size-based morphological variation between the sexes (*Frayer, 1980*; *Frayer & Wolpoff, 1985*). The wealth of ethnographic data on extant hunter-gatherers provides insights into the sexual division of foraging labor. Applying what we know about extant hunter-gatherer behavior to the bioarchaeological record, we can link musculoskeletal markers to habitual activities. Physical markers of activity include geometric variation in bone structure and function due to biomechanical loading (*Ruff, Holt & Trinkaus, 2006*), diaphyseal structure (*Bridges, 1989*; *Bridges, 1995*; *Ruff, 1987b*; *Ruff, 1987c*; *Schaffler et al., 1985*), degenerative joint disease (DJD), osteoarthritis, musculoskeletal markers (MSM) (*Eshed et al., 2004*), and dental wear patterns relative to tool manufacture (*Berbesque et al., 2012*; *Estalrrich & Rosas, 2015*). Our specific focus is on the relationship between biomechanics and bone functional adaptation in carpal and tarsal bones. Bone functional adaptation is driven by two general principles: first, organisms are able to structurally adapt to new living conditions; second, bone cells have capacity to respond to local mechanical stresses (*Ruff, Holt & Trinkaus, 2006*). We draw on the various markers of physical activity to inform our biomechanical approach and previous studies on other skeletal elements in the population of interest but limit our investigation to bone functional adaptation assessed by length-width ratios, which provide an index that can be used to compare relative bone strength (*Garn, 1972*; *Rauch, 2005*). Width is an indicator of relative bone strength (*Garn, 1972*; *Rauch, 2005*) because resistance to bending force is linked to bone diameter; bones functionally adapt to biomechanical stress by forming new bone on periosteal surfaces, which results in wider bones (*Macdonald, Hoy & McKay, 2013*). To the best of our knowledge, this paper is the first to examine the utility of carpal and tarsal metrics as evidence for bone functional adaptation to activity.

## Aims

The focus of this paper is to explore whether or not logistical mobility and domestic economies (subsistence and tool manufacture) are archaeologically visible in the feet (tarsals) and hands (carpals) and, if so, whether or not they reflect a sex-based division of labor. Specifically, we are interested in bone functional adaptation in response to biomechanical stress and use logged length-width ratios to assess relative bone strength (*Garn, 1972*; *Macdonald, Hoy & McKay, 2013*; *Rauch, 2005*). We also examine lateralization using raw measures for left and right sides. Much attention has been paid to sexual

dimorphism in carpal and tarsal bones in forensic contexts and with applications to sex identification bioarchaeology; these studies have had varied success with most pointing to the calcaneus and sometimes talus as the most diagnostic bones (*Bostanci, 1962*; *Bunning, 1964*; *Gualdi-Russo, 2007*; *Harris & Case, 2012*; *Hoover, 1997*; *Introna Jr et al., 1997*; *Kidd & Oxnard, 2002*; *Mastrangelo, De Luca & Sanchez-Mejorada, 2011a*; *Mastrangelo et al., 2011b*; *Riepert et al., 1996*; *Steele, 1970*; *Steele, 1976*; *Steele & McKern, 1969*)—for a review see (*Davies, Hackman & Black, 2014*). Here, we take a different approach by exploring whether activity is embodied in the often overlooked region of the hands and feet.

Carpals and tarsals have not been examined extensively in bioarchaeological contexts but are potentially very interesting bones. The hands and feet are heavily implicated in the daily activities of hunter-gatherers (e.g., mobility, use of weapons, tool-making, domestic economies) and that activity might be embodied in carpal and tarsal bones. Further, carpal and tarsal bones are less likely to be influenced by the noise created from conflicting signals of genetics and lifestyle that obfuscates differentiation of ultimate and proximate causes of variation that plagues the long bones (*Pearson Osbjorn, 2000*; *Ruff & Larsen, 2014*). Further, these dense and small bones tend to be among the better-preserved bones in archaeological contexts (*Henderson, 1987*; *Mann, 1981*) and, if daily activities are visible in these bones, we potentially capture data that might otherwise be lost in less well-preserved skeletons. At the very least, if they prove useful in identifying bone functional adaptation, they provide additional data for past activity reconstruction.

We examined carpal and tarsal bones from the Florida Early Archaic Windover Site. The Early Archaic is characterized by major climate change in North America and, along with it, a change in domestic economies. Warmer climate was driving big game north and broad-spectrum foraging was emerging as the primary subsistence economy, de-emphasizing the dietary contribution of males and increasing the contributions of females. The site consists entirely of the mortuary pond where mobile hunter-gatherers buried their dead (usually with grave goods, such as atlatls and stone tools). Due to the semi-tropical environment of Florida, seasonally occupied hunter-gatherer camps are not well-preserved and most of what we know of this period comes from mortuary ponds, rather than occupation sites. As one of the best-preserved and largest collections, Windover remains provide tremendous insights into this period. We generated research questions based on ethnographic and archaeological data from other hunter-gatherer populations and modified them based on specific patterns identified at the Windover Site in other studies on other skeletal elements. As such, the next section takes each area of interest (e.g., mobility) and starts with general patterns in hunter-gatherer populations then narrows down to what we know about Windover. Our specific research questions are placed at the end of Bioarchaeological Context after presenting the material that aided in their generation.

## BIOARCHAEOLOGICAL CONTEXT

### Mobility activity

#### *General hunter-gatherer patterns*

Hunter-gatherer mobility can be described as residential (moving camp to a new location as in seasonal occupation of resource rich areas) and logistical (individuals and/or smaller

groups temporarily split from the main group for shorter foraging trips or longer hunting trips) (*Binford, 1980*; *Kelly, 1983*)—this is particularly true for mobile hunter-gatherers that specialize in terrestrial resources (*Marlowe, 2005*; *Panter-Brick, 2002*; *Sahlins, 1968*). There is some evidence for sex-based variation in hunter-gatherer mobility; modern Hadza hunter-gatherer males engage in greater daily walking distances at faster speeds than females (*Berbesque et al., 2016*; *Raichlen et al., 2017*).

Distal limbs have been implicated to a greater degree than upper limbs in reflecting habitual activity due to the biomechanical forces arising from locomotive substrate (i.e., terrains on which activities are conducted), distances travelled in a day, and relative speed of locomotion (*Berbesque et al., 2016*; *Bridges, 1991*; *Bridges, 1995*; *Carlson, Grine & Pearson, 2007*; *Eshed et al., 2004*; *Lieverse et al., 2007*; *Malina & Little, 2008*; *Pearson Osbjorn, 2000*; *Pontzer et al., 2014*; *Raichlen et al., 2017*; *Ruff, 1987a*; *Ruff, 2000*; *Shaw & Stock, 2009*; *Stock, 2006*; *Venkataraman et al., 2013*; *Weiss, 2012*). Of interest to this study is how biomechanical stress from mobility and footedness might affect the tarsals—biomechanical stresses will cause the bone to functionally adapt to the stress through widening, as discussed previously (*Garn, 1972*; *Rauch, 2005*).

The ground reaction force generated by the bare (or minimally shod) foot contacting the ground is transmitted through the subtalar skeleton, with peak forces at heel-strike through the calcaneus and at heel-off through the metatarsophalangeal articulations (*Trinkaus & Shang, 2008*). Thus, during normal locomotion, the typical bipedal heel strike transmits body mass from the tibia to the rear foot (talus and calcaneus) to the ground (*Nordin & Frankel, 2012*) while the shock of impact is absorbed by the midfoot (navicular, cuboid, and cuneiform bones) (*Nordin & Frankel, 2012*). The calcaneus is most affected by the rear heel strike and calcaneal tuber length (a proxy for Achilles tendon moment arm length) is correlated with running economy (long calcaneal tuber = greater energy cost) (*Raichlen, Armstrong & Lieberman, 2011*) Most data from extant hunter-gatherers (*Hatala et al., 2013*; *Pontzer et al., 2014*) and barefoot populations suggest a rear heel strike (*Fredericks et al., 2015*) is preferred among experienced runners. Dorsal spurs on the calcaneus are linked to increased activity while plantar spurs are linked to standing, inactivity, and excess weight (*Weiss, 2012*). The navicular is a keystone bone in the arches of the foot that is impinged during foot strike by the talus and other cuneiforms. Structurally linking the rear- and mid-foot, it bears the transmission force of weight during the push-off phase of locomotion and experiences highly localized stress in middle one-third of the bone, which makes it prone to fracture in highly athletic individuals (*Coris & Lombardo, 2003*; *De Clercq, Bevernage & Leemrijse, 2008*; *Khan et al., 1994*; *Shakked, Walters & O'Malley, 2017*). Anatomically, the intermediate cuneiform articulates with the navicular proximally and second metatarsal distally. The second metatarsal-intermediate cuneiform joint is a highly stable keystone joint with limited mobility. Injury to the joint occurs via direct force from load applied to the base of the foot or indirect forces from a longitudinal force applied to a plantarflexed foot (*Liu et al., 1997*; *Rodgers, 1988*:1826), which is particularly pronounced in barefoot populations (*Franklin et al., 2015*; *Fredericks et al., 2015*; *Hollander et al., 2017*; *Hollander et al., 2016*; *Pontzer et al., 2014*).

While footedness in humans develops in late childhood (11–12 years old) with a right skew (*Gabbard, 1996*; *Gentry & Gabbard, 1995*), its influence on walking gait is not significant and is unlikely to affect the musculoskeletal system in the absence of other evidence of lateralization (*Zverev, 2006*). Thus, regions of the foot may be differentially shaped by daily logistical mobility that emphasizes either slow walking and stationary weight-bearing activity (such as might occur when foraging in a patch) or rapid locomotion such as brisk walking or running that requires greater shock absorption. And, if subsistence-based activities are assigned based on sex (*Frayer, 1980*; *Ruff, 1987b*), there may be sex-differences in how these regions of the feet vary. Ultimately, biomechanical stress will cause bone functional adaptation and increased width (*Garn, 1972*; *Rauch, 2005*).

### Windover patterns

The Windover bog, used seasonally for burials, was strategically located between the Indian River coastal lagoon system and the St. John's River—an area rich in marine, fresh water, and terrestrial resources—which indicates that the population did not have to travel long distances between seasons (*Adovasio, Soffer & Page, 2009*) and did not fission into smaller groups between visits to the pond (*Wentz, 2006*). Seasonal mobility is indicated by analysis of preserved stomach contents which were from plants and fruits maturing during July and October. In addition, growth ring data recovered from mortuary stakes indicated the wood was harvested in late summer/early fall (*Doran & Dickel, 1988a*). Ultimately, the residential mobility of the Windover population was limited to a constrained geographic area around the bog with most evidence pointing to emergence of sedentism (*Wentz, 2006*).

## Domestic economies and activity reconstruction
### General hunter-gatherer patterns

Domestic economies include subsistence activities and tool manufacture (which supports subsistence activities). Subsistence covaries with biological factors (e.g., habitat, reproduction, health) and cultural factors (e.g., social organization, sedentism, mobility). A comprehensive analysis of 229 hunter-gatherer diets, eco-environmental data, and plant-to-animal dietary ratios found that most populations consume similar amounts of carbohydrates (30–35% of the diet) except in more extreme environments (i.e., increases in desert and tropical grasslands and decreases in higher latitudes) (*Hiatt, 1978*). Indeed, there is a strong clinal pattern of variation in male and female caloric contributions to diet. Subsistence contributions by sex are inversely correlated with effective temperature, a combined measure of the intensity and annual distribution of solar radiation (*Bailey, 1960*; *Binford, 1980*). Higher latitudes and colder climates rely more on male caloric contributions from big game hunting while temperate and tropical regions rely more heavily on female caloric contributions across the spectrum (e.g., small game, fishing, and plants) (*Hiatt, 1978*). In general, males tend to increase foraging activities in more stable productive habitats (*Marlowe, 2007*) and females tend to decrease labor in subsistence activities when males are hunting big game (focusing instead on activities like weaving and cordage) (*Waguespack, 2005*).

Evidence for activity patterns is referred to as the 'holy grail' of bioarchaeology (*Jurmain et al., 2012*) due to the lack of written records in prehistory and the high amount of

inferential work involved to reconstruct them. In modern populations, however, tarsal bones are heavily implicated in repeated and sustained physical activity (e.g., running, walking long distances on a daily basis) (*Murray et al., 2006*). Sustained daily physical activity is consistent with the lifestyle described in ethnographies of contemporary hunter-gatherer populations (*Pontzer et al., 2015*; *Pontzer et al., 2014*; *Raichlen, Armstrong & Lieberman, 2011*; *Raichlen et al., 2017*). Likewise, carpal bones are responsive to physical stressors. Arthritis, among other bony changes, has been documented in modern populations and tied to repetitive tasks (e.g., painting, pipetting) (*El-Helaly, Balkhy & Vallenius, 2017*; *Heilskov-Hansen et al., 2016*). To date, there are no studies documenting bony or arthritic changes in the wrist due to foraging behaviors, but generally plant acquisition is thought to be repetitive and likely to employ some of the postures implicated in bony changes in the wrist (e.g., picking) (*Vignais, Weresch & Keir, 2016*).

Ethnographic data also indicate that other (non-foraging) domestic economies such as tool making, child care and carrying, butchering, food preparation, production of textiles, and carrying firewood and water create physical strain (*Bentley, 1985*; *Cowlishaw, 1981*; *Hurtado et al., 1985*; *Sahlins, 1968*). Increased reliance on tools is linked to evolutionarily more gracile bodies (*Trinkaus, 1983*) but tool manufacturing and use are detectable on the body through increased upper limb robusticity (*Carlson, Grine & Pearson, 2007*) and lateralization, or side preference (*Stock et al., 2013*). Unimanual activities (e.g., spear use) leave a distinct mark of directional asymmetry in the upper limbs compared to bimanual activities such as grinding or rowing (*Weiss, 2009*). Indeed, extant hunter-gatherers exhibit strong hand preference specifically when making and using tools (*Cavanagh et al., 2016*; *Robira et al., 2018*) which suggests a mosaic progression to the extreme lateralization we see in modern populations (*Stock et al., 2013*). And, there is evidence in the archaeological record of sidedness varying between the sexes (*Bridges, 1991*; *Bridges, 1994*). While domestic economies vary across groups, they tend to be sex-based and more frequently involve lateralized repetitive stress compared to subsistence and mobility (*Weiss, 2009*). Thus, bioarchaeological evidence of subsistence activities and tool manufacture may be found in repetitive stress to the musculoskeletal system and result in lateralized MSM, DJD, and osteoarthritis of the limbs involved. Carrying loads may place additional weight-bearing biomechanical stress on the foot and the domestic economies may serve to differentiate the wrists. These biomechanical stresses should differentiate bones more heavily implicated in specific activities and further differentiate between those more heavily engaged in those activities from those minimally (or not all) engaged in those activities.

## Windover patterns
### Subsistence economies

Paleodietary analysis from carbon and nitrogen bone-collagen values and archaeobotanical information suggest exploitation of riparian (river-based) resources rather than the more common Florida Archaic use of marine mammals or terrestrial fauna such as deer or rabbit (*Tuross et al., 1994*). Males and females did not have significantly different isotope values for major dietary components (*Wentz, 2006*; *Wentz et al., 2006*). Based on ethnographic data, the resource rich environment fostered by a milder wetter climate suggests greater

reliance on female caloric contributions to diet (*Hiatt, 1978*) and this is supported by grave goods—both males and females were found with materials for hunting small mammals, reptiles, and fish (*Hamlin, 2001*). There is evidence of some specialization by sex because only male graves contained atlatl components (typical of hunter-gatherers and male spear use (*Kelly, 1983*), spears, lithic projectiles, and hollow point awls (for making fishing nets) and only females and subadult graves contained direct evidence for food processing (e.g., butchered bone, mortar and pestle, containers) (*Adovasio et al., 2001*; *Hamlin, 2001*).

### Tool economies

Analysis of tools found amongst the grave goods suggests tool material choice (not type) was sex-based; females preferred shell and carapace, while males preferred antler bone (*Adovasio, Soffer & Page, 2009*; *Hamlin, 2001*). Few tool types were exclusive to one sex which suggests few activities were specific to one sex (*Hamlin, 2001*). Females also tended to have more decorative items (*Hamlin, 2001*) and were found exclusively with materials for plant-based medicine (*Adovasio, Soffer & Page, 2009*; *Hamlin, 2001*; *Tuross et al., 1994*). Interestingly, stone tools only play a cameo in the story of tools at Windover (*Adovasio, Soffer & Page, 2009*) while female goods (textiles, baskets, containers, medicines) have a starring role across the history of the pond. The absence of internment with stone tools suggests a cultural emphasis on the labor of women and products from both men and women in the domestic economy rather than an emphasis on male big-game hunting (*Adovasio, Soffer & Page, 2009*). This may be an outcome of climate change (*Doran & Dickel, 1988b*; *Milanich, 1994*) rapidly altering domestic roles from the Paleoindian to the Early Archaic periods.

### Activity reconstruction

DJD at Windover has been analyzed in two separate studies (*Smith, 2008*; *Wentz, 2010*), each using different but standard published methods. Smith used Waldron's method (*Smith, 2008*) based on bone eburnation (or polishing) (*Waldron, 1991*; *Waldron & Rogers, 1991*) and other arthritic changes at the joint (e.g., lipping, porosity). *Wentz (2010)* used the Western Hemisphere Health Index methods, a relative ordinal ranked scoring system in eight skeletal joints (*Steckel & Rose, 2002*). Both studies found high rates of DJD in males and females consistent with prehistoric hunter-gatherer lifestyles, but there were no statistically significant between-sex differences in DJD (*Smith, 2008*; *Wentz, 2010*). There were some sex-based trends that are relevant to the current study. First, DJD frequency in the cervical spine is particularly high in females and may be explained by food or palm leaf (for textile fibers) processing activities (*Wentz, 2010*) or carrying heavy loads (*Smith, 2008*) both of which are supported by grave good evidence (*Adovasio et al., 2001*; *Adovasio, Soffer & Page, 2009*; *Hamlin, 2001*). Males exhibit greater DJD in the lumbar region which suggests they were carrying heavy loads (perhaps game or goods during seasonal camp relocation) or stressed from repetitive motions related to hide processing (*Wentz, 2010*). Thus, both males and females may have been carrying heavy loads and both were engaging in similar or shared tasks (*Smith, 2008*: 45). Second, elbows were commonly affected which might be interpreted as male atlatl throwing but females exhibited more DJD in wrists, elbows and shoulders than males (*Wentz, 2010*) which suggests a shared activity
(*Smith, 2008*). Wrists exhibited little evidence of DJD (3/97 left and right wrists affected) but females had more hand trauma (*Wentz, 2010*) and males had a higher frequency of severe DJD in both hands (18% of the sample) (*Smith, 2008*). Again, similar or shared tasks are indicated. Third, there were some overall sex-based patterns in DJD with males exhibiting more knee and hip damage on the left and females exhibiting more severe change on the right (*Smith, 2008*), which might suggest footedness and increased mobility in males with more shock to the feet. Both sexes (37% of individuals) exhibited significant bilateral degeneration of talar-calcaneal articular facets (*Smith, 2008*), which might reflect high mobility and weight-bearing activities (*Weiss, 2012*), possibly running (*Franklin et al., 2015*; *Fredericks et al., 2015*; *Hollander et al., 2017*; *Raichlen, Armstrong & Lieberman, 2011*).

An examination of muscle insertion sites (*Hagaman, 2009*) found low levels of habitual stress (indicative of stressful repetitive activity) but muscle insertion sites were fairly robust indicating generally high activity patterns. As with the DJD results, there is much overlap between the sexes in scores further supporting the notion that most activities were shared. The lack of asymmetry in MSM, particularly in males due to the use of the atlatl, suggests a lack of repetition in this activity or other activities which exert symmetrical force on the upper limbs and obscure the lateralization of spear-throwing—possibly kayaking (*Hagaman, 2009*). But, the wrist is a complex system in which small changes in the anatomy of one bone be offset by changes in other aspects of the anatomy (*Maki, 2013*: 238). Analysis of fractures suggests interpersonal violence (affecting male crania and, less frequently, the post cranial skeleton) but the majority of trauma came from accidents with females slightly more affected than males (*Smith, 2003*). Ribs (often on the right side) were the most fractured in both sexes with ulnar fractures in second place. The vertebra exhibit evidence of compression fractures (more frequent in females and equal to ulna fractures in incidence) consistent with falls when landing in an upright position or carrying heavy loads (*Smith, 2003*). Overall, fracture patterns suggest accidents related to logistical mobility in the uneven intercoastal terrain and heavy underbrush along with heavy load carrying (*Smith, 2003*).

## MATERIALS

The Windover archaeological site (8BR46) is a National Historic Landmark dating to the Early Archaic Period (7500–5000 BCE) with calibrated radio-carbon dates from 9,000 to 7,929 BP. The site consists of a mortuary pond where seasonal mobile hunter-gatherers buried their dead. The Windover bog site (5,400 m$^2$) is one of a number of Florida Archaic Period 'wet cemeteries' or mortuary pond archaeological sites with underwater burials in peat; others include Little Salt Spring (8SO18) (*Clausen et al., 1979*), Republic Grove (8HR4) (*Wharton, Ballo & Hope, 1981*), and Bay West (8CR200) (*Beriault et al., 1981*). Little is known about this time period because skeletal remains in these sites are most often from a very small number of individuals, are often very fragmentary, and some sites were not excavated systematically (such as Little Salt Springs) (*Wentz & Gifford, 2007*). Further, the semi-tropical conditions of Florida are less than ideal preservation conditions, especially for temporary occupation sites that might consist of minimal non-organic materials.

In general, the North American Archaic Period (8000 to 1000 BC) is characterized by hunting-gathering subsistence economies with dietary staples including nuts, seeds, and shellfish (*Milanich, 1994*). The Florida Archaic Period follows the same pattern (e.g., broad spectrum hunting, fishing, and plant gathering and use of freshwater resources) with increased exploitation of coastal shellfish and marine resources. The comparatively wetter climate (*Halligan et al., 2016*) created an abundance of resources and subsistence strategies were no longer dominated by big game. The broad spectrum foraging strategy that emerged is reflected in more complex tool kits (*Doran & Dickel, 1988b*; *Milanich, 1994*). See *Brown (1994)*, *Klingle (2006)* and *Milanich (1994)* for overviews of Florida prehistory.

The Windover site was used as a mortuary pond for 5–6 short periods of activity, peaking at 7,450 BP (*Doran & Dickel, 1988a*; *Doran & Dickel, 1988b*). Burials furthest from the pond edge at time of excavation dated to the earliest period of mortuary pond use and those closest, more recent. Roughly 100 burials were undisturbed with fully articulated bones; ages ranged from infancy to over sixty-five, with 52% classified as subadults (*Purdy, 1991*). Most individuals were buried within 24 to 48 h after death (*Doran & Dickel, 1988a*) in a flexed position, on the left side with heads oriented to the west, and pinned by sharpened stakes approximately 1m below the surface of the peat (*Hauswirth, Dickel & Lawlor, 1994*). The nearly neutral pH of the pond (6.1–6.9) created ideal conditions for preservation of both skeletal and soft tissues; allowing researchers to sequence DNA from preserved brain matter (*Hauswirth, Dickel & Lawlor, 1994*), reconstruct diet from preserved stomach contents (*Tuross et al., 1994*), and study textile industries (*Adovasio et al., 2001*). The population exhibited predominantly good health and included individuals of extremely advanced age (50+) for hunter-gatherer groups which reflects local resource abundance (*Klingle, 2006*) and medical practices (*Adovasio, Soffer & Page, 2009*; *Hamlin, 2001*; *Smith, 2003*; *Tuross et al., 1994*; *Wentz, 2006*). Common to hunter-gatherer populations, adults of both sexes exhibited a high incidence of osteoarthritis (*Smith, 2008*), frequent enamel defects (*Berbesque & Doran, 2008*; *Berbesque & Hoover, 2018*), and skeletal trauma (*Smith, 2003*). Overall, female health was worse than male health (*Wentz, 2006*; *Wentz et al., 2006*).

## Research questions
Mobility
1. Previous research indicates that male hunter-gatherers walk more and at greater speeds than females. Windover data suggest a shared labor load, reduced emphasis on big game hunting, and evidence for heavy load carrying in both sexes. Which model is reflected in bone functional adaptation? We might expect the shared load model at Windover to result in no significant between-sex differences in rear foot tarsal variables that reflect the ground force reaction during locomotion and midfoot tarsal variables that absorb shock during locomotion.
2. The asymmetrical DJD in lower limbs and fracture patterns reviewed previously suggest there may be tarsal asymmetry. Is this evidenced by directional asymmetry in tarsal bones?

Subsistence

1. Previous research finds greater hand trauma and domestic economy production in females but greater DJD in males. Are there between-sex differences in carpal bones?
2. Based on findings of asymmetry in contemporary populations (hunter-gatherer and industrialized) engaged in repetitive manual tasks, we might expect prehistoric hunter-gatherers regularly engaged in repetitive manual tasks (food processing, tool production) to exhibit a similar pattern. Is there directional asymmetry in the carpal bones studied?

## METHODS

### Raw data collection and variables

Carpal and tarsal bones with standard anatomical reference points intact were included only from adults who had well-defined features used in sex assessment (given the focus on sex-based morphological variation). Sex assessment was carried out by *Doran & Dickel (1988a)* using nonmetric pelvic and cranial traits and metric analysis of femoral and humeral head dimensions following standard osteological methods (*Buikstra & Ubelaker, 1994*). The final sample was 44 (27 males, 17 females) but sample size varies by measurement. All measurements were taken on right and left sides, when available for inclusion in the directional asymmetry analysis. Length and width of whole bones were measured in millimeters for four carpals (capitate, hamate, lunate and scaphoid) and four tarsals (calcaneus, intermediate cuneiform, navicular, talus). Length and width of talar (trochlea) and calcaneal (load arm) articular surfaces were also measured. See Table 1 for measurement details using standard anatomical landmarks.

### Size difference data and directional asymmetry analysis

Directional asymmetry is assessed by comparing side differences (here, selected right and left hand and foot bones). Confounding factors for directional asymmetry analysis are sexual dimorphism and trait size variation, discussed in-depth in asymmetry methodology papers that were applied to this analysis (*Palmer, 1994*; *Palmer & Strobeck, 1986*; *Palmer & Strobeck, 2003*). Preliminary raw data inspection via scatterplots and outlier statistical tests (e.g., Grubb's statistic) were performed to eliminate confounding effects of outliers (data recording, trait size, anomalous individuals) (*Palmer & Strobeck, 2003*). Univariate analysis generated input for a mixed model ANOVA (sides × individuals) which was performed in the Fluctuating Asymmetry Calculations Worksheet (V.11) (*Palmer, 1994*; *Palmer & Strobeck, 1986*; *Palmer & Strobeck, 2003*).

### Bone functional adaptation, index data and analysis

As discussed previously, bone functional adaptation in response to biomechanical forces can be assessed by bone width as a proxy for relative bone strength (*Garn, 1972*; *Rauch, 2005*). But, sexual dimorphism (*Fairbairn, 1997*; *Jungers, 1984*; *Smith & Cheverud, 2002*) and trait size variation (*Huxley & Tessier, 1936*; *Lewontin, 1966*) confound analysis conducted on raw width measurements. For example, a comparison between the width of the intermediate cuneiform and the width of the talus simply demonstrates that the talus is a wider bone

**Table 1  Description of measurements on carpal and tarsal bones.**

| Variable | Orientation | Description |
|---|---|---|
| Scaphoid length | proximal | scaphoid tubercle-lateral most point |
| Scaphoid width | palmar | bisection of scaphoid ridge |
| Capitate length | medial palmer | proximal-distal end |
| Capitate width | lateral | thinnest point |
| Lunate length | proximal | proximal-distal end |
| Lunate width | proximal | medial-lateral sides |
| Hamate length | lateral | proximal end-distal ridge (between metacarpal facets) |
| Hamate width | lateral | most medial to most lateral side of the facet |
| Calcaneus length[a] | lateral | most posterior point of tuberosity to most anterior-superior point of cuboidal facet[b] |
| Calcaneus width[a] | superior | minimum horizontal width through body taken anterior to the tuberosity and posterior to talar posterior facet[b] |
| Calcaneus load arm length[a] | superior | most posterior point of talar posterior articular surface to most anterior-superior point of cuboidal facet[b] |
| Calcaneus load arm width[a] | superior | most lateral point of posterior articular surface to most medial point of sustentaculum tali[b] |
| Talus length[a] | superior | flexor hallucis longis muscle sulcus at posterior aspect of talus to most anterior point on articular surface for navicular[b] |
| Talus width[a] | superior | most lateral point of articular surface for lateral malleolus to opposite point of tibial articular surface[b] |
| Trochlea of the talus length[a] | superior | anterior-posterior plane[b] |
| Trochlea of the talus width[a] | superior | perpendicular to projected line for maximum length of the trochlea[b] |
| Navicular length[c] | distal | medial tuberosity to lateral cuneiform facet |
| Navicular width[c] | inferior | between intermediate and medial cuneiform facets |
| Int. cuneiform length[c] | superior | proximal and distal midpoint |
| Int. cuneiform width[c] | superior | thickest middle portion |

**Notes.**
[a]Weight-bearing.
[b]Steele and Bramblett, 1988.
[c]Shock-absorbing.

because it is a larger bone. Likewise, a comparison between male and female talus widths simply demonstrates that males tend to be larger than females. An approach using raw measurements does not further our understanding of meaningful trait differences beyond absolute size.

A common resolution to the effect from sexual dimorphism is to take a ratio of two variables, such as length and width (*Fairbairn, 1997*; *Huxley & Tessier, 1936*; *Jungers, 1984*; *Jungers, Falsetti & Wall, 1995*; *Lewontin, 1966*; *Mobb & Wood, 1977*; *Smith & Cheverud, 2002*). We take length and width because width is an indicator of relative bone strength (*Garn, 1972*; *Rauch, 2005*)—and length and width are commonly taken, easily replicated measurements. As previously mentioned, resistance to bending force is related to bone diameter because apposition of new bone on periosteal surfaces (a functional adaptation to biomechanical stress) widens the bone (*Macdonald, Hoy & McKay, 2013*). Thus, the ratio of length to width provides a useful index of relative bone strength and is a marker of biomechanical forces acting on bone functional adaptation from activity. An index of

one describes a bone that has equal length and width (1:1). A ratio of greater than one describes a bone that is longer than it is wide and a ratio of less than one describes a bone that is wider than it is long. When comparing bones that have indices that lie exclusively either above or below the set point of one, the relative bone strength can be inferred by smaller numbers. For example, if all the bones are wider than they are long (a ratio value below 1), those with the smallest values are relatively wider than those with larger values (but, see below for interpreting logged values).

The issue of trait size variation is common in asymmetry studies (*Palmer, 1994*; *Palmer & Strobeck, 1986*; *Palmer & Strobeck, 2003*) and often resolved by scaling data to the natural log after taking the absolute value of the length to width index (*Mobb & Wood, 1977*; *Palmer & Strobeck, 2003*). The natural log creates a symmetric and homoscedastic dataset that retains the original linear scale of standard deviation to the mean (the spread or variation of the data) (*Sokal & Rohlf, 1995*). We applied this final transformation to our data and that resulted in a total of 10 index variables (capitate, hamate, lunate, scaphoid, calcaneus, calcaneus load arm, intermediate cuneiform, navicular, talus, talus trochlea tibia) to test bone functional adaptation analysis. We transformed left and right sides separately because each represents an individual measure of biomechanical stress. Equation (1) represents the full transformation of raw length and width variables to the final index that accounts for the confounding issues of sexual dimorphism and trait size. The index is derived from the natural log ($\log_n$) of the absolute value of length-to-width ratio ($L$:$W$) for each trait ($x$)

$$\log_n |x(L:W)|. \tag{1}$$

An example of how this index mitigates the confounding effects of sex-based and trait-based size differences in bone metrics is seen in (Fig. 1 and Figs. S1–S9). The left panel of Fig. 1 is a scatterplot of raw measurements for the navicular. The right panel of Fig. 1 is a scatterplot of the index values. The distribution of raw data on the left reflects the absolute differences in body size (males larger than females) while the right panel reflects real differences once absolute size differences are eliminated by application of Eq. (1). Our index values describe bones that are absolutely longer than they are wide excepting the calcaneus load arm which is wider (than it is long) and the lunate which is equal in length and width (14.43:14.90 or .99). See Table 2 for an average of ratio values using raw length and width measurements (L:W) and an average of index values using Eq. (1). Table 2 demonstrates that the linear scale of the original data is retained and can be interpreted in a similar manner as before with the exception that values greater than zero are longer than they are wide (i.e., relatively long) and values less than zero are wider than they are long (i.e., relatively wide). To return to the earlier example, if all the bones are wider than they are long (a logged ratio value below 0), those with the smallest values (larger negative values) are relatively wider than those with larger values (closer to zero).

Tests for between-group differences using index variables characterized differences in relative width as an indication of biomechanical stress acting on the area of interest. The General Linear Model (GLM) was used to test for between-sex differences in tarsal and carpal variables. All results were evaluated relative to confidence intervals, power, and estimated effect size.

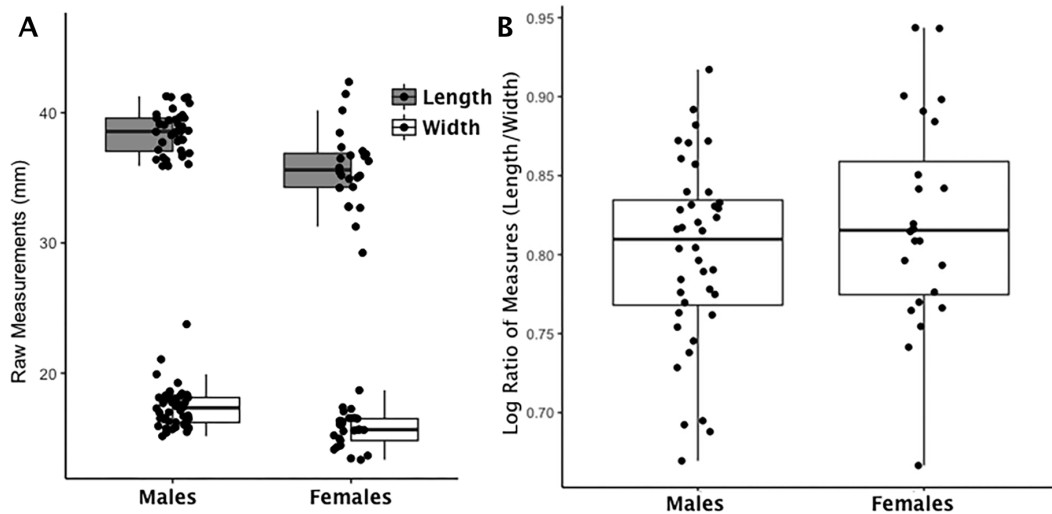

**Figure 1** **Boxplot of navicular raw length and width measurements by sex (A); Boxplot of navicular index values by sex (B).** (A) Each data point for the navicular represents an individual length (shaded boxplot) or width value (unshaded boxplot); males and females are displayed separately. (B) Each data point for the navicular represents an individual logged ratio index value; males and females are displayed separately.

**Table 2** **Mean per variable for Ratio (L:W) and Index (logged L:W) variables.**

|  | Ratio value | Index value |
|---|---|---|
| Calcaneus Load Arm | 0.78 | −0.25 |
| Lunate | 0.99 | −0.03 |
| Talar-Trochlea | 1.02 | 0.02 |
| Intermediate Cuneiform | 1.08 | 0.08 |
| Talus | 1.23 | 0.21 |
| Hamate | 1.32 | 0.27 |
| Capitate | 1.85 | 0.61 |
| Navicular | 2.25 | 0.81 |
| Scaphoid | 2.47 | 0.90 |
| Calcaneus | 3.12 | 1.13 |

## RESULTS

Outlier data identified by Grubb's statistic were examined relative to other measurements for the individual and were removed if inconsistent and not a result of data entry error (see notes tab in the raw data file). Five outliers were found to be due to data entry error and were corrected (see notes tab in the raw data file). All variables were normally distributed (Table S1). Descriptive data for all variables by sex are found in Table S2.

Are there between-sex differences in tarsal weight-bearing index values and tarsal shock-absorbing index values? As discussed in the introduction, body weight is born by the rear foot (talus and calcaneus) during heel strike (*Nordin & Frankel, 2012*; *Trinkaus & Shang, 2008*) and impact shock is absorbed by the midfoot (navicular, cuboid, and cuneiform
**Table 3  Univariate results from GLM for tarsal variables.**

| | Test | F | df | Sig | $\eta p$ | Power | Sex | Mean | SE | n | 95% CI | |
| --- | --- | --- | --- | --- | --- | --- | --- | --- | --- | --- | --- | --- |
| | | | | | | | | | | | Lower | Upper |
| Weight-Bearing | Calcaneus Load Arm | 0.20 | 1 | 0.66 | 0.00 | 0.07 | Male | −0.25 | 0.01 | 28 | −0.28 | −0.224 |
| | | | | | | | Female | −0.24 | 0.02 | 18 | −0.28 | −0.207 |
| | Calcaneus | 5.70 | 1 | 0.02 | 0.12 | 0.65 | Male | 1.15 | 0.01 | 28 | 1.13 | 1.177 |
| | | | | | | | Female | 1.11 | 0.02 | 14 | 1.08 | 1.139 |
| | Talus | 0.03 | 1 | 0.88 | 0.00 | 0.05 | Male | 0.21 | 0.01 | 28 | 0.19 | 0.236 |
| | | | | | | | Female | 0.21 | 0.01 | 14 | 0.18 | 0.239 |
| | Trochlea | 0.00 | 1 | 0.96 | 0.00 | 0.05 | Male | 0.01 | 0.01 | 28 | −0.01 | 0.033 |
| | | | | | | | Female | 0.01 | 0.01 | 14 | −0.02 | 0.04 |
| Shock-Absorbing | Intermediate Cuneiform | 3.77 | 1 | 0.06 | 0.07 | 0.48 | Male | 0.07 | 0.01 | 34 | 0.05 | 0.086 |
| | | | | | | | Female | 0.10 | 0.01 | 20 | 0.07 | 0.126 |
| | Navicular | 1.12 | 1 | 0.30 | 0.02 | 0.18 | Male | 0.80 | 0.01 | 34 | 0.78 | 0.822 |
| | | | | | | | Female | 0.82 | 0.01 | 20 | 0.79 | 0.847 |

bones) (*Nordin & Frankel, 2012*). We used a multivariate GLM to assess sex differences and test the hypothesis that Windover had a shared labor load that deviates from what might be expected in typical hunter-gatherers with a sex-based division of labor. Results of the multivariate GLM using index values (the index corrects for sex dimorphism and trait size variation) confirmed expectations: there were no significant between-sex differences in weight-bearing (Pillai's Trace $F = 1.522$, $df = 4$, $p = 0.205$, $\eta p = 0.132$; observed power $= 0.437$) or shock-absorbing tarsals (Pillai's Trace $F = 2.599$, $df = 2$, $p = 0.084$, $\eta p = 0.093$; observed power $= 0.495$). Both tests are underpowered with small effect sizes. The *p*-value for weight-bearing bones is high, which suggests minimal chance of Type II error. The *p*-value for shock-absorbing bones, however, is low which might indicate Type II error from the small sample size. Between-subjects tests for each index variable (Table 3, Figs. 2 and 3, Figs. S1–S5) indicate that the calcaneus exhibits a sex-based trend, but the *p*-value (0.021) is not significant after multiple-hypothesis test correction ($\alpha = 0.0125$).

Is there directional asymmetry that might indicate lateralization of the foot evidenced in tarsals as suggested by asymmetrical DJD in lower limbs and fracture patterns? The mixed-model results did not indicate significant between-side differences in raw tarsal measurement variables (length and width for four bones and length and width for two articular facets), which suggests there is no directional asymmetry or 'footedness' present (Table S3).

Is there a between-sex difference in carpal bones based on previous research showing greater hand trauma and domestic economy production in females? Or, is the previous research showing greater DJD in male hands an equalizing effect on the skeletal embodiment of different activity patterns? The *p*-values for the results of a multivariate GLM on carpal index variables for between-sex differences were not significant (Pillai's Trace $F = 2.811$, $p = 0.059$, $\eta p = 0.0398$; observed power $= 0.639$). The *p*-value exceeds the set $\alpha$ level (0.05) by a very narrow margin (0.09) and the test was underpowered (64%), both of which suggest possible Type II error due to insufficient sample size (the female sample was much smaller

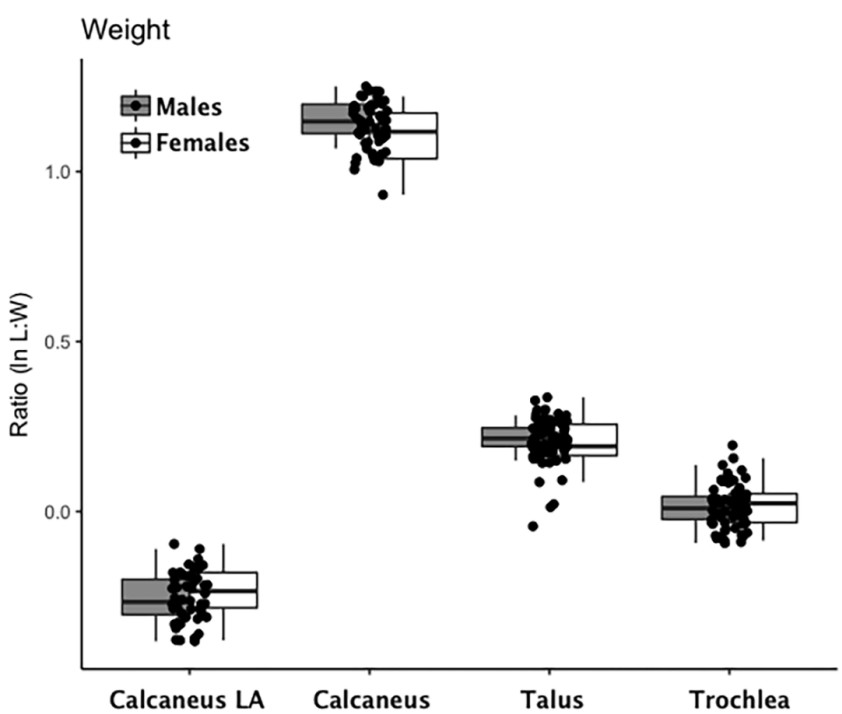

**Figure 2** **Boxplot of weight-bearing tarsal variables (talus, calcaneus) by sex.** Each data point represents an individual index value (*y*-axis) for each of the four weight-bearing index variables on the *x*-axis. Male boxplots are shaded; female boxplots are not shaded.

**Table 4** **Univariate results from GLM for carpal variables.**

| | Levene | Sig | F | df | Sig | $\eta p$ | Power | | Mean | SE | n | 95% CI Lower | Upper |
|---|---|---|---|---|---|---|---|---|---|---|---|---|---|
| Capitate | 3.22 | 0.09 | 2.85 | 1 | 0.11 | 0.13 | 0.36 | M | 0.60 | 0.02 | 28 | 0.56 | 0.63 |
| | | | | | | | | F | 0.65 | 0.02 | 21 | 0.60 | 0.70 |
| Hamate | 2.15 | 0.16 | 1.11 | 1 | 0.31 | 0.05 | 0.17 | M | 0.28 | 0.02 | 21 | 0.24 | 0.32 |
| | | | | | | | | F | 0.25 | 0.02 | 14 | 0.20 | 0.30 |
| Lunate | 4.32 | 0.05 | 2.71 | 1 | 0.12 | 0.12 | 0.35 | M | −0.08 | 0.03 | 27 | −0.15 | −0.01 |
| | | | | | | | | F | 0.01 | 0.04 | 20 | −0.08 | 0.10 |
| Scaphoid | 0.22 | 0.64 | 1.10 | 1 | 0.31 | 0.05 | 0.17 | M | 0.85 | 0.03 | 22 | 0.79 | 0.92 |
| | | | | | | | | F | 0.91 | 0.04 | 19 | 0.82 | 1.00 |

than the male sample)—an unfortunate problem common to many bioarchaeological studies. Between-subjects tests for each index variable (Table 4, Figs. S6–S9) indicate no significant between-sex differences for any variable.

Is there directional asymmetry that might indicate lateralization of the hand evidenced in carpal bones as suggested by handedness in complex tasks for making tools and textiles? The mixed model results did not indicate significant between side differences for raw carpal measurement variables (length and width for four carpal bones), which suggests there is no directional asymmetry or 'handedness' present (Table S3).

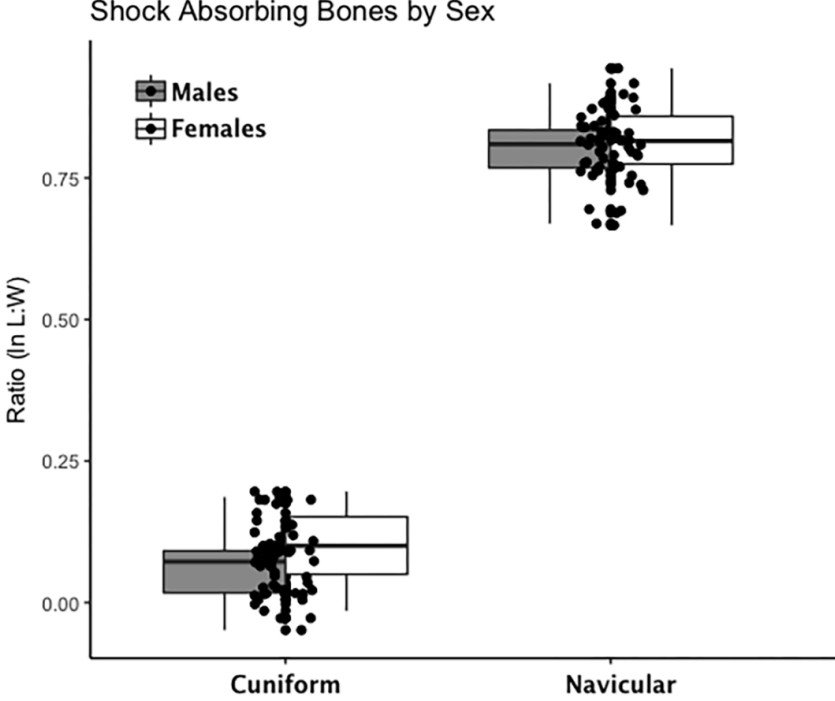

**Figure 3** **Boxplot of shock-absorbing tarsal variables (navicular, intermediate cuneiform) by sex.** Each data point represents an individual index value (*y*-axis) for two shock-bearing index variables on the *x*-axis. Male boxplots are shaded; female boxplots are not shaded.

## DISCUSSION

Archaeological data reconstructing the environment surrounding the mortuary pond suggest that the larger region was rich with riparian and terrestrial resources due to climate warming at the start of the Early Archaic. As megafauna distributions increased in latitude, Windover subsistence strategies progressively relied less on big-game hunting and more on broad spectrum foraging (*Doran & Dickel, 1988a*; *Halligan et al., 2016*). Grave good analysis and activity reconstructions suggest heavy overlap between the sexes in domestic economies rather than a strict sex-based division of labor (*Hagaman, 2009*; *Hamlin, 2001*; *Smith, 2008*; *Smith, 2003*; *Wentz, 2006*).

### Tarsal variation

Based on previous analysis, we speculated whether the biomechanical stress of load carrying might influence bone functional adaptation in the tarsals. Typical hunter-gatherer males have faster walking speeds and tend to walk greater distances than females (*Pontzer et al., 2014*). If males were more specialized in hunting activities that required them to walk longer distances at faster speeds than females, we might expect relatively wider (i.e., stronger) weight-bearing bones in males (lower index values for talus and calcaneus variables)—this expectation is predicated on body weight being born by the rear foot (talus and calcaneus) (*Nordin & Frankel, 2012*; *Trinkaus & Shang, 2008*). We also might expect males to have greater impact shock from locomotion, which is absorbed by the midfoot (navicular,

cuboid, and cuneiform bones) (*Nordin & Frankel, 2012*). The archaeological interpretation of the division of labor at Windover, while limited to grave goods, suggests a shared model of labor. If males are less specialized and share domestic economy tasks with females, we would expect no differences in the sexes and this is what we found. That said, both tests are underpowered and there may well be significant differences in the shock-absorbing bones that were not identified due to small sample sizes per variable. Still, the effect is very low (in this study), which suggests that any differences that might exist are not likely to have biological significance.

The calcaneus index values are the only ones that exhibit a clear sex-based trend (even if not significant after correction for multiple hypothesis testing). Recall that the bipedal heel strike during locomotion transmits body mass from the tibia to the rear foot (talus and calcaneus) to the ground (*Nordin & Frankel, 2012*). Extant Hadza hunter-gatherers tend to favor a midfoot strike (*Pontzer et al., 2014*) while the Daasanach pastoralists favor a rearfoot strike (*Hatala et al., 2013*), the latter of which significantly impacts the calcaneus more so than other bones and which increases dorsal spurs on the calcaneus (*Weiss, 2012*). Calcaneal dorsal spurs are correlated with running economy (long calcaneal tuber = greater energy cost) (*Raichlen, Armstrong & Lieberman, 2011*)—this has been noted in endurance runners in Kenya who appear to favor a forefoot strike (*Lieberman et al., 2015*). Because the Windover population was experiencing a comparatively wetter climate than in previous archaeological periods (*Halligan et al., 2016*), big game was harder to find. We know that the population had an abundance of local resources and emphasized broad spectrum foraging over big-game hunting (*Doran & Dickel, 1988b*; *Milanich, 1994*). Male burials, however, sometimes are associated with atlatls which suggests some big game hunting and might explain why males have relatively longer calcaneus bones than females (higher index values, closer to zero, are interpreted as relatively longer and lower index values, larger negative numbers, are interpreted as relatively wider). While some research argues that barefootedness causes wider feet (*Lieberman, 2013*), the situation is more complex. Clinical evidence of the effect on the foot due to barefoot walking and running during growth and development suggests that while the forefoot does widen (*Franklin et al., 2015*), the overall length is increased at the expense of width even when controlling for potentially confounding effects of demographic and developmental variation in activity and body weight (*Hollander et al., 2017*). A large cohort ($n = 520$) of children aged 6–18 in South Africa (habitually barefoot, even in school) and Germany showed that the wider foot cedes to a longer foot in the South African barefoot cohort. Variation in published findings may be related to factors such as the confounding factors influencing adult foot morphology and, in the one other study conducted on habitually barefoot children (*D'AoÛt et al., 2009*), factors such as BMI and ligament tensility (*Hollander et al., 2017*). The story of foot morphology and barefootedness is not yet complete.

## Foot lateralization

Despite evidence suggesting footedness would not be visible in the archaeological record (*Zverev, 2006*), bioarchaeological evidence suggested there might be—namely, lateralization in the lower limbs might have resulted in lateralization of tarsal bones with the dominant

foot having larger mid foot widths (due to biomechanical pressure to strengthen the bone during impact) or rear foot widths (due to biomechanical pressure to strengthen the bone during weight-bearing). Specifically, males had greater DJD on both knees and the left talus and calcaneus (*Smith, 2008*:47). These lateral patterns did not translate to differences in the tarsals bones in this population. As previously discussed, footedness in humans develops in late childhood (11–12 years old) with a right skew (*Gabbard, 1996*; *Gentry & Gabbard, 1995*) but its influence on walking gait is not significant and not likely to affect the musculoskeletal system in the absence of other evidence of lateralization (*Zverev, 2006*). A future study might confirm this by examining bone functional adaptation in response to foot preference in populations with known physical activity patterns that evidence clear lateralization. Further, the hallux has been identified as a potentially significant bone that might be examined in bioarchaeological contexts due to its involvement in the extreme plantarflexion associated with barefoot locomotion (*Franklin et al., 2015*).

## Carpal variation

We postulated we would find between-sex differences in carpal bones due to increased hand trauma (*Wentz, 2010*) and cervical vertebra DJD in females (possibly linked to food and textile processing activities) (*Adovasio et al., 2001*; *Adovasio, Soffer & Page, 2009*; *Wentz, 2010*) compared to males who had a higher frequency of severe DJD in both hands (18% of the sample) (*Smith, 2008*). The multivariate test conducted on the index variables for the four carpal bones did not result in a significant finding of difference between males and females but, as noted in Results, the test was underpowered and the *p*-value was only slightly higher than the significance threshold. Thus, a Type II error may have occurred. The female carpal sample is smaller than the male carpal sample and there is no way to predict what the differences might actually be. The between-subjects tests did not indicate any individual variable was significantly different between the sexes but these were underpowered due to small sample size, particularly for females and the hamate index. Given that the female sample is much smaller than the male sample, we cannot comfortably make any conclusions relative to the research question. We can, however, suggest that there is very small (if any) effect size from activity marking on the bones due to biomechanical stress (in this study). And we note that a lack of difference circumstantially supports prior analysis suggesting heavy overlap between the sexes in labor as evidenced by male graves containing tools for domestic labor (*Hagaman, 2009*; *Hamlin, 2001*; *Smith, 2008*; *Smith, 2003*; *Wentz, 2006*).

There were some interesting trends worth noting even if they are not part of statistical hypothesis testing. First, female carpals (excepting the scaphoid) exhibit a greater range of index values than males (Table 4, Table S2). Females engage in broader non-diet activity (e.g., weaving, cordage) when male contributions to diet are larger (*Waguespack, 2005*). Given that the Early Archaic is characterized by rapid change from big game hunting to broad spectrum foraging (*Doran & Dickel, 1988a*; *Halligan et al., 2016*; *Milanich, 1994*) and given that Windover females are engaging in both subsistence and non-subsistence domestic activities (*Hagaman, 2009*; *Hamlin, 2001*; *Wentz, 2010*), perhaps our data reflect this transition and greater variety in female workload. First, female specific activities include

health care as evidenced by grave goods for medicine preparation (*Hamlin, 2001*; *Wentz & Gifford, 2007*) and the Windover population experienced a variety of ailments (e.g., post-fracture bone alignment, surviving childhood stressors) (*Wentz, 2006*; *Wentz, 2010*; *Wentz et al., 2006*). Second, the sex-based preference for tool materials suggests a gendered ideology surrounding tasks (*Hamlin, 2001*). Finally, a between-sex health comparison found females had poorer health, which might be attributable to greater life stress from a heavier workload (*Wentz et al., 2006*). Thus, while we cannot conclude there are differences in biomechanical stress on carpal bones, we can suggest there may be evidence of females engaging in a wider variety of tasks. Ultimately and partly as a result of the very small effect size, the carpals do not appear to be useful (at least in this study) in identifying between-sex differences in biomechanical stressors from activity. But, perhaps the results reflect the 'shared-load' model put forward for Windover division of labor.

**Hand lateralization**

Our final area of inquiry was whether there was any evidence for lateralization in the carpal bones; this was based on previous research that suggested that hunter-gatherers exhibit strong handedness when engaged in complex tasks (*Cavanagh et al., 2016*; *Robira et al., 2018*; *Stock et al., 2013*). The complex tool kit at Windover would have provided an avenue for handedness to be archaeologically visible, but neither male nor female carpal bones exhibited significant directional asymmetry that would suggest lateralization. We might have expected some lateralization in males if the carpals were implicated in spear-throwing and males were engaged in heavy hunting activities but the Windover archaeological record does not suggest this was the case. Further, the bioarchaeological record shows that DJD and MSM patterns were shared between the sexes which suggests that males were not likely regularly engaged in hunting or the markers of habitual spear-throwing were offset by changes in other aspects of wrist anatomy (*Maki, 2013*: 238). A previous study on a population with clear handedness identified only two bones exhibiting directional asymmetry, the lunate and trapezium (*Reina et al., 2017*) which suggests handedness is not likely to be archaeologically visible.

# CONCLUSION

This paper explores the biomechanical stresses acting on bone functional morphology in carpal and tarsal bones as a novel method of identifying the embodiment of logistical mobility and domestic economies (subsistence and tool manufacture). We identified key characteristics of hunter-gatherer mobility and domestic economies that might leave their mark on the bone. And, we used the wealth of data published on the Windover population from other bioarchaeological and archaeological studies to guide our expectations of what we might find in the carpals and tarsals.

We were particularly interested in the archaeological visibility of lateralization. Previous studies on the subject of lateralization in humans (past and modern) are not entirely on agreement on what expectations might be. Modern populations exhibit footedness (*Gabbard, 1996*; *Gentry & Gabbard, 1995*) and handedness (*Stock et al., 2013*) but extant hunter-gatherers only exhibit handedness during activities involving complex tasks

(*Cavanagh et al., 2016*; *Hurtado et al., 1985*; *Robira et al., 2018*) and footedness has not been studied. Even though hunter-gatherers have been studied in terms of gait and locomotion (*Fredericks et al., 2015*; *Hatala et al., 2013*; *Pontzer et al., 2014* #5479; *Lieberman et al., 2010*; *Niemitz, 2010*; *Pontzer et al., 2014*), studies on modern populations indicate that the influence of walking gait on footedness is not significant (*Zverev, 2006*). We used raw measurements of length and width in carpal and tarsal bones to identify any directional asymmetry in the sample (as a proxy for lateralization). There was no evidence of directional asymmetry in the sample and our findings seem in line with the general understanding that hunter-gatherers do not exhibit hand preference enough for handedness to be archaeological visible. Further, hunter-gatherers, even if they exhibit foot preference (as suggested by lateralization of musculoskeletal markers), foot preference has little impact on the bones.

We were also interested in whether there were sex-based differences in weight-bearing regions of rear foot (talus and calcaneus) and shock-absorbing regions of the mid foot (intermediate cuneiform and navicular). We used an index variable for four tarsal bones and two tarsal articular surfaces to examine sex-based differences in mobility. While most hunter-gatherer populations might be expected to vary between the sexes based on different mobility patterns, Windover hunter-gatherers were not expected to because the archaeological evidence for labor suggests a shared load. We found no sex-based differences in either area of the foot which may support the shared load model given that male bones are not showing a significant bone functional adaptation to greater locomotory biomechanical stress (walking longer distances at greater speeds). But, the tests were underpowered due to small sample sizes and female samples were smaller than male samples. Still, the effect size was so small that any significant differences would not likely have biological significance. Male calcaneus index values are higher (closer to zero on the logged scale) which suggests they are relatively longer than female index values. The result is not statistically significant after correcting for multiple hypothesis testing but it might signify some remaining male hunting activity, or at the very least, some minimal between-sex differences in mobility (*Franklin et al., 2015*; *Fredericks et al., 2015*; *Hollander et al., 2017*).

Finally, we asked if carpal metrics supported the shared load model (*Hamlin, 2001*) or the partially shared load model with greater burden on the part of females (*Wentz, 2006*; *Wentz, 2010*). We used an index variable for four carpal bones to examine differences in activity (shared workload or sex-based division of labor) (*Cavanagh et al., 2016*; *Robira et al., 2018*). Again, the tests were underpowered due to small sample sizes and female samples were smaller than male samples. The female range of index values is greater than the male and we suggest that perhaps females were engaging in a wider variety of tasks than males; there is some bioarchaeological evidence to support this postulation in the model for partially shared (male and female) tasks and heavier female workloads (*Wentz, 2006*; *Wentz, 2010*). Ultimately, we note that (in our sample at least), the effect size of activity pattern in differentiating the sexes is too small to be captured without a sufficiently large sample size. If this is generally true (i.e., the same pattern is found in other collections), carpals are not a useful proxy for sex-based activity reconstruction in the absence of other traditional indicators.

While we cannot infer too deeply from the site because it is a mortuary pond and only reflects internment ritual, we can argue that our findings muddy the water in terms of the shared load model. Males and females are engaged in similar mobility patterns that emphasize weight-bearing rather than shock-absorbing activity. But, there is some evidence for sex-based differences in mobility—males are potentially still ranging further afield than females in pursuit of increasingly rare big game or simply have different locomotive patterns than females. While there are no sex-based differences in carpal bones, greater variation in female index values suggest that females may have been more specialized to specific tasks, or some engaged in female specific activities (as suggested by previous assessments of the high value placed on female labor at Windover) (*Adovasio, Soffer & Page, 2009*). But, if our findings are typical of how bone is marked by activity, the effect size of sex-based difference is too small to be of great use in interpreting activity patterns at archaeological sites. Finally, we can conclude that lateralization in the wrist and foot is not archaeologically visible in this population.

## ACKNOWLEDGEMENTS

We wish to acknowledge Dr. Frank L. Williams (Georgia State University) for suggesting the frame of the study and Dr. Lia Betti (University of Roehampton) for reading an earlier version of this manuscript. We also thank Professor Glen Doran for access to the collection and his mentorship and support throughout our careers.

### Funding
The authors received no funding for this work.

### Competing Interests
Kara C. Hoover is an Academic Editor for PeerJ.

### Author Contributions
- Kara C. Hoover conceived and designed the experiments, performed the experiments, analyzed the data, prepared figures and/or tables, authored or reviewed drafts of the paper, approved the final draft, collected data.
- J. Colette Berbesque analyzed the data, prepared figures and/or tables, authored or reviewed drafts of the paper, approved the final draft.

### Data Availability
Hoover, Kara (2018): Hoover and Berbesque-raw data.xlsx. figshare. Dataset. https://doi.org/10.6084/m9.figshare.5977375.v3.

### Supplemental Information
Supplemental information for this article can be found online at http://dx.doi.org/10.7717/peerj.5564#supplemental-information.

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
