# Peer review of "Early Holocene morphological variation in hunter-gatherer hands and feet"

_PeerJ, doi:10.7717/peerj.5564_

## Round 0.1 · original submission · Major Revisions

Dear authors,

I have received comments from three reviewers on your manuscript. In general, the reviewers consider your study to be interesting and potentially suitable for publication in PeerJ following revision. One of the main issues that has popped up with the reviewers is that the manuscript would benefit greatly from revision of the text to improve clarity. This is especially the case for the methods, where reviewers have requested improvements to the descriptions of the indices (possibly augmented by figures), and have raised a series of comments and concerns regarding the protocol used (Reviewer #2 and #3). Please address these points, and provide precise details for the method steps, to help the reader to trace clearly which analyses were conducted (and on which set of bones). Reviewer #3 has also requested some further justification regarding the use of tarsals and carpals, and has made some recommendations regarding ordination and interpretation of the indices. Please also provide the raw data as a supplementary file, as per requirements for publication in PeerJ.

·

Basic reporting

Data Check: PeerJ requires that the raw data is available. The current version of the supplementrary tables contains summary statistics and a variety of details for other statistical analyses but not the raw measurements and indices. Please inclue a data file containing them.

Everything else looks good here.

Experimental design

Please see my comments to the authors. The key points relevant to experimental design are:

The most important criticism I can offer is that the measurements taken and indices calculated should be clearly described (and probably illustrated to make them as clear as possible) within the text. It was unclear what, exactly they were.

Second, please describe (perhaps following line 305 in the current text) which variables were used to characterize “shock absorbing” bones and which were used for “weight bearing” bones and provide a brief justification.

Validity of the findings

Please see my comments to the authors

Additional comments

This manuscript presents an interesting and innovative analysis of the dimensions and proportions of hand and foot bones of male and female hunter-gatherers from the early through mid-Holocene site of Windover in Florida. The authors present a very nice summary of the site’s cultural context and relevant data from studies of modern hunter gatherers on variation in patterns of sexual division of labor. The authors report a larger than expected range of variation in females and higher averages for some indices in females than males, contrary to expectations. Little sign of lateralization is apparent in the Windover hands or feet. They interpret these findings as evidence of a heavier activity load in females. Overall, this manuscript is a useful and stimulating contribution. A few details should be clarified before it is accepted.

Data Check: PeerJ requires that the raw data is available. The current version of the supplementrary tables contains summary statistics and a variety of details for other statistical analyses but not the raw measurements and indices. Please inclue a data file containing them.

The most important criticism I can offer is that the measurements taken and indices calculated should be clearly described (and probably illustrated to make them as clear as possible) within the text. It was unclear what, exactly they were.

Second, please describe (perhaps following line 305 in the current text) which variables were used to characterize “shock absorbing” bones and which were used for “weight bearing” bones and provide a brief justification.

Around lines 374-378 and line 408, more caution on the causes of DJD is warranted. It has a complex etiology, sometimes traceable to differences in activity, but sometimes not. Please see Jurmain et al. (2012 – see below) for an overview.

In the paragraph under the header “Carpal Variation” (line 413 and following), please be more clear if absolute size or the values for an index are being discussed. The text usually implies absolute sizes are under discussion and this may be misleading.

More minor critiques:

Around the current lines 298-302, please note what software package(s) were used to analyze the data.

Line 435 – word choice: perhaps “intensive foraging” rather than “intense foraging”?

Reference cited:

Jurmain R, Alves Cardoso F, Henderson C, Villotte S. 2012. Bioarchaeology’s holy grail: the reconstruction of activity, in A.L. Grauer (ed.) A Companion to Paleopathology: 531-552. Crichester, West Sussex: Wiley-Blackwell

Reviewer 2 ·

Basic reporting

The writing could be more clear and unambiguous. For example, it is not always clear if all carpals or tarsals are being used or only the most diagnostic elements. The literature cited is relevant, however several of the references are not particularly specific to the archaeological context under discussion, or are a little dated. The figures are difficult to interpret and need more detail on what data is being presented.

Experimental design

No information is provided regarding ethical protocols, therefore it is not clear if ethical standards were followed. The methods could be more clear to allow for replication. Additionally, the authors could better clarify why this is the best possible method for this particular investigation.

Validity of the findings

Some of the statistical conclusions are difficult to interpret and could be presented with greater clarity. Speculation is evident in the conclusions but could be more clearly identified as such. As this method is not widely used, and the context is not ideal for inferring past lifeways and how they affect morphology, the authors could be more cautious with their interpretations.

Additional comments

In line 28, the bracket following tarsal should be closed.

In line 29, what is the nature of the differences that suggest Windover occupants engaged in heavy impact mobility.

In several cases, such as line 45, citations or comments that belong within one set of brackets are shared between multiple sequential bracketed texts.

Ethnographic research is referenced several times in this study broadly, are there any ethnographic or historic records specifically pertinent to this group from this region? If there are, please provide further details, and if not, that could be clearly stated.

The term listed on lines 47-48 is conventionally ‘musculoskeletal markers’ or ‘musculoskeletal stress markers.’

Throughout the text and starting on line 71, the author refers to the tarsals collectively and does not always clarify which tarsals, in particular, are under discussion. As they outline that different tarsals have particular functional contributions, why refer to them collectively and not consistently identify the tarsals under investigation or discussion?

In line 87, the foot may be differentially shaped by daily logistical mobility?

The authors reference Hiatt (1978) in line 100, are there more recent studies that look into these proportions? Additionally, is there any research specific to the region under investigation in this present study?

In lines 132 and 149, are the authors referring to all tarsals and carpals?

In lines 134, the authors note that forensic studies and some bioarchaeological research have investigated size variation in carpals and tarsals? Are there particular studies that have looked at variation between groups known to have different activity patterns? Is there any clinical research on this topic? If so, it should be mentioned specifically. If not, and this study is the first to undertake this approach, this should be explicitly addressed. Additionally, it should be acknowledged that it is being tested on a group without known activities, which may present some interpretive issues.

In line 140, the authors note that these bones are heavily implicated in daily activities. Are there other studies demonstrating that this is the case among prehistoric groups? Is there direct evidence suggesting they may be more diagnostic of activity than long bones, which are frequently used in investigations of past activity patterns.

In line 144, there are more recent reviews and critiques of long bone variation than Pearson (2000). The authors could include additional references on this topic if they are aiming to demonstrate that carpals and tarsals are better proxies than other conventionally used bones.

In line 147, carpals and tarsals may be more frequently recovered in poorly preserved skeletons. The authors could note that such research could be more broadly applied after further testing in groups with either known activities or strong archaeological or ethnographic evidence of likely daily actions.

In line 171, a definition of what a mortuary pond is and what it conventionally contains would be helpful. This is not clearly stated in the methods section. Also, if the materials are complete, why not cross-reference across the skeleton using different methods?

In line 210, only male graves contain atlatl?

On lines 231-232, what are the different methods used by Smith and Wentz to look at DJD? Additionally, were there no limitations to these studies? Were the sample sizes large? Were only adults analysed? Also, the bones analysed are not outlined, except when results are mentioned. Were whole skeletons analysed? Or only axial elements and long bones?

In line 282, was skeletal sex assessment conducted as part of this study? If so, what methods were used? If not, are sex assessments from previously published reports? If measurements were taken on both sides to allow for directional asymmetry measurements, was one side biased in the comparisons between units? Also, why was Grubb’s statistic used in particular?

In line 289, in addition to sexual dimorphism and trait size variation, could additional factors like physiological stresses or diet be impacting size variation in this group?

Throughout the text, for example on line 325, the authors could reiterate the bones in question when referring to shock-absorbing bones.

In line 339, what do the authors mean by potentially significant differences? What are the criteria used in this paper for significance testing? If a particular criteria (i.e. a=0.05) was used to determine significant differences, please indicate, and be consistent when describing when results are significant or not.

In line 400, is it possible that the greater range of variation in females may relate to the larger sample size? Also, future avenues of research might involve looking at these variables in known populations or further investigate reaction norms in carpals and tarsals in response to activity?

In line 417, increased DJD in what elements?

In line 451, variation may relate to female tasks, but it is worth reiterating that this is may only be the case once factors related to size are taken into account.

In line 462, is the wrist particularly implicated in spear throwing? Studies that have looked at living subjects engaging in spear throwing seem to predominantly implicate the shoulder. Why are carpals a better proxy for such activity than, for example, the shoulder girdle? Also, handedness has not been previously observed in the carpals, a) why was it expected in this study, and b) is it possible that insufficient force is felt at the carpals in particular to reflect handedness? This could be explored in more detail.

In line 468, it would be helpful if the aims of the study were restated before being summarized.

In line 473, if females experienced worse health, would this not potentially indicate some degree of inequality between males and females?

In line 502, what ethical approval process was used to undertake this study? Are there any institutions to be acknowledged?

In the figures in the tables, listing the carpals and tarsals involved in the calculations would be extremely helpful to the reader. For figure 1, are the labels of left and right correct? Also, is the Y-axis the mean of log-transformed ratio data? What is the measure or what are the units involved with this data? These graphs are not informative without more details on what the numbers used actually represent. In Figure 2, why is there a line connecting males and females? This indicates some direct or transformative relationship. Also, if the axes represent a similar measure, the axes should be numerically aligned. In Figure 3, the icons used to identify the different carpals used (also, if these are the only carpals used this should be clarified earlier in the study) are extremely similar and difficult to distinguish. Other images like a circle or a square would be easier to identify.

If Windover is a mortuary pond, and as described in the conclusion, does not have an associated archaeological site, are the skeletons recovered from the pond best described as occupants? Additionally, what greater contextual information from other sites in this region have informed hypotheses and lifestyle inferences? These details could be clarified in greater detail.

Reviewer 3 ·

Basic reporting

The main goal of the presented articles to differentiating the mobility and subsistence patterns between males and females in the prehistoric hunter-gatherers from North America. In general, because of abundance of local resources studied population was limited to explore large geographic area. The grave goods demonstrate tool choice based on the sex. Both women and man participated in the domestic economy. DJD studies of Windover population clearly demonstrate separation of labor or different activity patterns between males and females. Difference between left and right DJD frequency also was demonstrated in the previous studies. This study greatly benefits if authors could directly compare DJD studies results.
I cannot judge English unfortunately
I recommend remove or shorten introduction before “Aim” and move up in introduction material section and bland into Windover population mortality, mobility, Subsistence economies… with evidence of another hunter-gatherer population. It allows authors to demonstrate Windover population difference from other hunter-gatherer groups and to define unique aspect why this population will be important to study labor and activity difference between male and female. Authors are providing the research questions, but they do not provide the predictions on the base of subsistence strategy of this population and why these predictions are made and what are their arguments.
Tables and figures are fine. Authors only share ratio and not raw data.
Authors choose carpal and tarsal bone for following reason (1) because both tarsals and carpals heavily implicated to the activities; (2) they are less influenced by genetic factors (3) carpals and tarsal are well preserved in archeological samples. All these arguments can be good for any upper and lower limb bone as well. It is not clearly justified why carpals and tarsals are not influenced by genetic factors, also in this particular population it not necessary to study carpal and tarsal in order do not lose the information about mobility patters when they have the most complete skeletons of hunter-gatherers.
Even DJD demonstrate site specific difference between male and females, it does not exclude that both males and females are caring heavy weight and are mobile in the defined geological area. Present work greatly benefits if authors can indicate how they can distinguish on the tarsal bones L versus W mobility related difference if any. How tarsal measurements are correlated to AP versus ML diameters of long bone mid-shaft?

Experimental design

Authors would like to see on bone length and breadth ratio how it is differentiating labor difference as well as heavy load carrying and high mobility or running. They are using log transform length-to-width ratio for each tarsal and carpal bones. When ratio is used most of time it is difficult to distinguish which of two variable affects to the value of ratio. I strongly recommend using bi-variate plot for L versus W measurements to explore real difference between male and female as well as right and left. I also do not convince that variable that are chosen are correct to differentiate load bearing sites. I recommend using each articular surface diameter to calculate surface area for each bone and scale of same bone product L and W or use body weight and compare them between sex and left and right. This will show if compression forces acting on the carpal bone during different manipulation of the hand.
For tarsal, it is unclear how authors can demonstrate shock-absorption using L/W measurement of bone. Probably subchondral porosity or microdamage will be best indicator is one would like to distinguish between shock-absorption difference in males and females. Please describe how one can distinguish on the one bone (not complex structure) weight bearing versus shock-absorption? And, please provide literature supporting this.

Validity of the findings

Previous DJD study of same population demonstrate sex difference of affected site of skeleton on the hand bones. This strongly demonstrate division of labor as well as difference of heavy load caring difference between different sex. I expected if measurements for this study are chosen correctly the result needed to be paired previous results of DJD of same population.
https://academic.oup.com/rheumatology/article/44/4/521/1774828
https://www.ncbi.nlm.nih.gov/pmc/articles/PMC3266544/

After plotting Log transformed mean ration and SD for each bone (See file attached). Means are almost identical. SD in female is larger when male, but patters for each bone is similar. Presented result do not support any activity or mobility difference, simply because they are sampling general bone size. Here it will be interesting to see body weight distribution of this population in the male and female pool.

Additional comments

line 28 (4 carpal and 6 tarsal. Change (4 carpal and 6 tarsal).
Line 31 “shock-absorbing bones” I do not understand what does it means and how one can depict shock-absorbing on bone using size of bones? If they can describe cartilage degeneration and subchondral porosity or microdamage, but presented data do not support this. In addition, the maximum absorbed energy of subchondral bone is approximately 4 times lower when cartilage (https://www.sciencedirect.com/science/article/pii/S1751616113001628?via%3Dihub#f0020)
Line 45 (Frayer & Wolpoff 1985) (Frayer 1980). Change to (Frayer & Wolpoff 1985; Frayer 1980).
Line 46 ascribed to each of the sexes change to ascribed status to each of the sexes
Line 208 after (Hiatt 1978) put space and eliminate one dash line
And many others

Annotated reviews are not available for download in order to protect the identity of reviewers who chose to remain anonymous.

---

## Round 0.2 · Minor Revisions

A reviewer who previously reviewed your manuscript has commented positively that the revised version has addressed the concerns raised in the previous round of review and that the manuscript should be accepted for publication. The reviewer has made a very minor suggestion, and I agree, that the authors should consider slightly expanding their text on the observations of foot morphology in unshod locomotion. Following consideration of this minor point, I am happy to accept the manuscript for publication.

·

Basic reporting

The paper meets PeerJ's criteria for Basic Reporting: the language used is clear and unambiguous; sufficient references and background are provided; the article has a clear, professional structure, and the raw data is provided; the relevant results are provided for the hypotheses.

Experimental design

All of this is OK.

Validity of the findings

All of this looks OK.

Additional comments

This review concerns the revised version of "Early Holocene morphological variation in hunter-gatherer hands and feet" (#26634) by K. C. Hoover and J. C. Berbesque. The paper presents an interesting analysis of whether lateralization in hand and foot bones reflects work loads at the early Holocene site of Windover in Florida. The authors have done a good job of addressing the criticisms I had of the first version of the paper. I urge the Editor of PeerJ to ACCEPT the revised version or to ACCEPT it after MINOR REVISIONS.

The final revision I can suggest concerns the text on lines 524-526 of the revised manuscript in which the authors summarize the effects of barefoot walking/running during development on the shape of the foot and its arches (longer & narrower feet with lower arches as a result of barefoot running). Please be aware that Daniel Lieberman has argued (with supporting data) precisely the opposite - that barefoot walking & running leads to wider feet (although perhaps much of this is ligament rather than bone) and higher arches. (Lieberman, D. 2013. The Story of the Human Body: Evolution, Health, and Disease. New York: Vintage.). As a result, please double check these facts (or simply delete them since they are somewhat tangential to the data in hand).

---

## Round 0.3 · accepted · Accept

Dear authors, I'm happy to accept your revised manuscript, thank you for addressing the very minor point raised in the last round of review. I fully agree with your comment regarding the complexity of interactions at play in relation to foot morphology and biomechanics of shod/unshod locomotion. I look forward to seeing your paper published.

#